# Optimization of Transpedicular Electrode Insertion for Electroporation-Based Treatments of Vertebral Tumors

**DOI:** 10.3390/cancers14215412

**Published:** 2022-11-02

**Authors:** Helena Cindrič, Damijan Miklavčič, Francois H. Cornelis, Bor Kos

**Affiliations:** 1Faculty of Electrical Engineering, University of Ljubljana, 1000 Ljubljana, Slovenia; 2Memorial Sloan-Kettering Cancer Center, New York, NY 10065, USA

**Keywords:** treatment planning, numerical modeling, bone tumors, tumor treatment, minimally invasive treatment

## Abstract

**Simple Summary:**

Electroporation has sparked great interest regarding its use in medicine. When planning electroporation-based treatments, the main goal is to determine the best possible electrode position and voltage amplitude that will ensure treatment of the entire target tissue’s volume. However, this process is still mainly performed manually or using computationally intensive genetic algorithms. This study presents an algorithm for optimizing electrode positions based on spatial information of the electric field distribution in the target tissue. The algorithm is currently designed for the electrochemotherapy of vertebral tumors via a transpedicular approach but could be adapted to other anatomic sites in the future. The algorithm performs successfully for different spinal segments, tumor sizes, and locations within the vertebra. Application of the algorithm significantly reduces the time and expertise required to create a treatment plan for the electrochemotherapy of vertebral tumors.

**Abstract:**

Electroporation-based treatments such as electrochemotherapy and irreversible electroporation ablation have sparked interest with respect to their use in medicine. Treatment planning involves determining the best possible electrode positions and voltage amplitudes to ensure treatment of the entire clinical target volume (CTV). This process is mainly performed manually or with computationally intensive genetic algorithms. In this study, an algorithm was developed to optimize electrode positions for the electrochemotherapy of vertebral tumors without using computationally intensive methods. The algorithm considers the electric field distribution in the CTV, identifies undertreated areas, and uses this information to iteratively shift the electrodes from their initial positions to cover the entire CTV. The algorithm performs successfully for different spinal segments, tumor sizes, and positions within the vertebra. The average optimization time was 71 s with an average of 4.9 iterations performed. The algorithm significantly reduces the time and expertise required to create a treatment plan for vertebral tumors. This study serves as a proof of concept that electrode positions can be determined (semi-)automatically based on the spatial information of the electric field distribution in the target tissue. The algorithm is currently designed for the electrochemotherapy of vertebral tumors via a transpedicular approach but could be adapted for other anatomic sites in the future.

## 1. Introduction

Electroporation is a phenomenon in which short high voltage electric pulses are used to change the structural integrity of the cell membrane and consequently increase the membrane permeability. Depending on the pulse parameters, the phenomenon can be either reversible, meaning that the cells remain unchanged in the long term, or irreversible, meaning that the cells cannot recover from the changes in the membrane and eventually die [1,2,3,4]. Both reversible and irreversible electroporation have sparked great interest regarding their use in medicine. Reversible electroporation can be used in combination with chemotherapeutic agents, a treatment known as electrochemotherapy (ECT) [5,6,7,8], or with genetic material, a treatment known as gene electrotransfer (GET) [9,10]. Irreversible electroporation (IRE) has emerged as a promising alternative to thermal methods for the ablation of tumors and soft tissues [11,12,13,14,15,16,17]. It is generally accepted that electroporation occurs in tissues at a specific electric field strength, i.e., the electroporation threshold. A complete coverage of the target tissue volume with an electric field above a certain value is required to achieve a therapeutic effect in all electroporation-based treatments [18,19].

When planning electroporation-based treatments, the main goal is to determine the best possible electrode position and applied voltage amplitude that will ensure the electroporation of the clinical target volume (CTV) and cause minimal damage to the surrounding healthy tissue [19,20,21]. The most common method for predicting the outcome of an electroporation-based treatment is to apply a tissue-specific threshold to the computed electric field distribution and determine the coverage of the CTV, i.e., to calculate the fraction of the target volume covered by an electric field strength of at least the (tissue-specific) threshold value [19,22,23,24]. However, by calculating the fraction of the CTV above the threshold, the computation results are reduced to a single numerical value, and the spatial information about the distribution and local strength of the electric field that the calculation provides is lost.

The determination of the optimal electrode positions is still mainly performed by hand. Most attempts to optimize electrode positions and voltages are based on either genetic algorithms (GA) or parametric sweeps [25,26,27], which are time-consuming and require significant computational power. Moreover, the criteria used for optimization are mainly the fraction of CTV coverage and the total volume of damaged healthy tissue, thus failing to exploit the valuable spatial information provided by the computation.

In this study, we present an algorithm for the optimization of electrode positions for the electrochemotherapy of vertebral tumors without using computationally intensive genetic algorithms. The developed algorithm considers the electric field distribution in the target tissue, identifies the regions not covered by a sufficiently high electric field (i.e., undertreated regions) in the CTV, and uses this information to iteratively move the electrodes from their initial positions to their final positions to cover the whole CTV. The applied voltage is also adjusted by the optimization process. Technological constraints such as ensuring the appropriate electrode spacing and accounting for the limitations (e.g., maximum current and voltage) of commercially available pulse generators are also considered by the algorithm.

The algorithm is developed for the treatment of vertebral tumors using two needle electrodes inserted through the pedicles into the vertebral body. The concept is based on our previous studies on the electrochemotherapy of spinal metastases using a transpedicular approach [28,29]. However, the algorithm is designed according to the modular principle and can be adapted to other anatomic sites in the future by adding new “modules” and reusing some of the existing ones. The algorithm’s source code and all models of the vertebral tumors created in this study are available in an open database at https://doi.org/10.6084/m9.figshare.21270111.v1 (accessed on 5 October 2022).

## 2. Materials and Methods

### 2.1. Dataset Preparation

To construct anatomical models of vertebrae, six lumbar and six thoracic vertebrae (of the lower thoracic region T8–T12) were segmented from medical images of three patients using Slicer 3D [30] and Mimics 24.0 (Materialise NV, Leuven, Belgium). First, a threshold was applied to the medical image to obtain a rough mask of the bone, which was then sliced into individual vertebral masks, smoothed, and manually corrected. The individual vertebral masks were saved as surface meshes and further smoothed and uniformly re-meshed using the 3-matric 16.0 (Materialise NV, Leuven, Belgium).

Each completed vertebral mesh was imported into COMSOL Multiphysics (COMSOL Inc., Stockholm, Sweden), where a model of a spherical tumor was inserted at one of the three different locations in the vertebral body: central position (Figure 1a), anterior-lateral position (Figure 1b), and posterior-inferior position (Figure 1c). Each tumor was modelled with three different radii: 5 mm, 7.5 mm, and 10 mm. Therefore, 9 different tumor models were created for each of the 12 vertebrae, resulting in a total of 108 models, which served as the dataset for the computation. A block was built around each vertebral model representing the surrounding healthy tissue. Two needle electrodes were added to each model, modelled in COMSOL as cylinders with a fixed radius of 0.6 mm and exposure length of 20 mm.

A mesh convergence study was performed in COMSOL to ensure that the discretization error was minimal. A physics-controlled mesh was used, and the element size varied from “extremely fine” to “normal”. The volume of tissue exceeding 400 V/cm, CTV coverage, and computation time were evaluated at each mesh size and compared to the extremely fine mesh. Using the “fine” mesh size resulted in no change in CTV coverage, a 1% change in total electroporated tissue volume, and a 96% decrease in computational cost, so it was selected as the final mesh size.

The block of healthy tissue also served as the boundary of the computational domain, and convergence with respect to the size of the boundary block was assessed. The size of the boundary block was gradually decreased in 5 mm increments in every spatial direction, until the relative change in the electroporated tissue volume surpassed 1% between two successive boundary block sizes. The CTV coverage remained unaffected at all tested sizes. The final size of the boundary block was 130 × 120 × 75 mm, which was the same in all models.

### 2.2. Computational Approach

COMSOL Multiphysics, a finite element analysis software, was used for the computations of electric field distribution in the models. MATLAB (MathWorks, Natick, MA, USA) via LiveLink was used to control the model’s setup and solution. The electrode positions and orientations are determined by the algorithm in each iteration, and the electrode parameters in COMSOL geometry are corrected accordingly. The applied voltage is also determined by the algorithm and corrected accordingly in the model. The electric field distribution is computed independently in each iteration with new electrode positions.

The electric field distribution *E* in the target tissue is computed indirectly by solving the partial differential equation for electric potential *V* (Equation (1)) for stationary conditions, governed by the equation:(1)$$\begin{array}{c} \nabla \cdot \left(\sigma \left(-\nabla V\right)\right)\;=\;0, \end{array}$$
where *σ* is the tissue conductivity, and *V* is the electric potential. The external domain boundaries are set as electrically insulating. The increase in tissue conductivity due to local electric field (Equation (2)) is modelled with a smoothed Heaviside function (with continuous second derivative), which is defined for each tissue separately. Thus, the conductivity in eq. 1 becomes a function of the local electric field:(2)$$\begin{array}{c} \sigma \to \sigma \left(\left|E\right|\right).\; \end{array}$$

The parameters of the smoothed Heaviside function for each modelled tissue are listed in Table 1. A detailed description of the computational approach can be found in previous studies [28,31].

### 2.3. Algorithm Structure

The algorithm was developed entirely in MATLAB, but the computations are performed in COMSOL and connected to MATLAB via LiveLink (see Section 2.2 Computational Approach). From the entire model dataset, 12 models were randomly selected, involving 4 samples of tumors in each radius studied. This group of models was used as the “training set” for the development of the algorithm.

The algorithm was developed through computational experimentation with the training set; a series of evaluations was performed and analyzed to obtain the best overall performance. The algorithm iteratively changes the positions of two electrodes within the vertebral body. The goal is to achieve complete coverage of the clinical target volume with a sufficiently high electric field in as few iterations as possible. The flowchart of the algorithm’s structure is shown in Figure 2, while the algorithm’s structure is explained in detail in Section 2.3.1, Section 2.3.2, Section 2.3.3, Section 2.3.4, Section 2.3.5.

#### 2.3.1. Clinical Target Volume Coverage

The boundary between gross tumor volume (GTV—i.e., total tumor volume as seen on medical imaging) and healthy tissue is usually not sharp; the presence of tumor cells outside the tumor volume depends on the tumor’s type and growth pattern. Therefore, in clinical practice, it is common to treat not only the GTV but also a margin of healthy tissue (5–10 mm), i.e., a safety margin, around the tumor volume to treat possible tumor cells or micrometastases. The tumor, together with the safety margin, forms the so-called clinical target volume (CTV).

For safety reasons, the thresholds for electroporation used in practice are generally quite high, and sometimes it may be difficult to cover the entire CTV for larger tumor radii. However, if the safety margin is taken into account, very few or no cells are expected at the outer edge of the CTV; therefore, the so-called soft coverage of the CTV is introduced in this study, in which the threshold for electroporation at the outer edge of the CTV is not strictly enforced. For this purpose, a weighting map of the CTV was created, where each voxel in the map is assigned a weight based on how far the voxel is from the boundary of the GTV. The GTV has a weight of 1, which means that it must be covered (at least) by the threshold electric field. In the safety margin, the weighting decreases linearly and reaches zero outside the CTV. The weighting map can be easily adapted to the tumor type. For example, in metastatic tumors, a higher weight (e.g., 0.5) can be assigned to the outer edge of the CTV to provide additional safety.

When calculating the center of mass of undertreated areas of the CTV (below the desired threshold), the weighting map is taken into account; therefore, the resulting center of mass is closer to the GTV boundary than to the outer boundary of the CTV. In the current implementation, the safety margin for all tumor radii is set at 5 mm. The margin can be easily adjusted (e.g., 10 mm) for different tumor types. An example of the CTV’s weighting map is shown in Figure 3.

#### 2.3.2. Input and Initialization

The operator must identify two points in the CT scan of the vertebra for each pedicle: the entry point, positioned in the narrowest part of the pedicle, and a second point that indicates the pedicle’s orientation/direction, as shown on Figure 4a–c. The position of each electrode is determined with the coordinates of the electrode tip and the direction vector pointing from the electrode tip to the entry point. The operator-selected points determine the starting position of the electrode, with the second point serving as the electrode tip (Figure 4d). Throughout the optimization, the entry point remains fixed to ensure transpedicular insertion, while the electrode tip is iteratively changed by the algorithm. For this reason, the entry point must be chosen with care.

At the beginning of the optimization process, the electrode’s geometry is pulled towards the tumor’s center of mass (Figure 4e)—as described in Section 2.3.3, Optimization of Electrode Positions—to compute the initial field distribution. This initialization step significantly decreases the solution time, compared to using the starting positions, selected by the operator.

#### 2.3.3. Optimization of Electrode Positions

The algorithm is constructed in a modular form, with different “forces” acting on the electrodes. The final force acting on the electrode, and, therefore, effectively moving the electrode to a new position, is a weighted sum of all acting forces. In the current implementation, three main processes controlling the electrode positions are considered: attractive force toward the tumor’s center of mass (CoM), attractive force toward the undertreated areas of the CTV, and the repulsive force between the electrodes, maintaining appropriate inter-electrode distance to prevent short-circuit. In the future, the algorithm can be adapted for other anatomic sites by adding new forces and reusing some of the existing ones.

##### Attractive Force to Tumor’s Center of Mass

By seemingly connecting the point of the first electrode to the rear of the second electrode and vice versa, two lines are obtained. The line segment that is the shortest distance between the lines is calculated; the point in the middle of this line segment is considered the center of electrode geometry (illustrated on Figure 5a). The force *F_geo_* is the vector pointing from the center of the electrode geometry to the CoM of the tumor. The point of application of *F_geo_* is at the tip of the electrode.

##### Attractive Forces to Undertreated Regions of the Clinical Target Volume

The areas of the CTV where the local electric field strength does not reach the threshold for electroporation (400 V/cm) are considered undertreated regions. The undertreated region is often disconnected, resulting in n undertreated “islands” within the CTV. The islands that are less than 10% of the size of the largest island are discarded and the CoM of each remaining island is calculated. Note that while computing the CoM of the island, the weighting map of the CTV is considered as well (see Section 2.3.1. Clinical Target Volume Coverage). The distance from each electrode rear and tip to CoM of each island I (illustrated on Figure 5b) is calculated. The forces acting on the electrode tip (*F_i,T_*) or rear (*F_i,R_*) are defined to mimic the gravitation force:(3)$$\begin{array}{c} F_{i,T}\;=\;\frac{m_{i}}{{\left\|d_{i,T}\right\|}^{2}}\cdot \frac{d_{i,T}}{\left\|d_{i,T}\right\|},\;i\;=\;1,\ldots ,n\; \end{array}$$
(4)$$\begin{array}{c} F_{i,R}\;=\;\frac{m_{i}}{{\left\|d_{i,R}\right\|}^{2}}\cdot \frac{d_{i,R}}{\left\|d_{i,R}\right\|},\;i\;=\;1,\ldots ,n \end{array}$$
where *m_i_* is the mass of the *i*-th island, calculated as the weighted number of voxels in the island; *d_i,T_* and *d_i,R_* are the vectors from electrode tip (*T*) or rear (*R*) to the *i*-th islands CoM. The application points of forces *F_i,T_* and *F_i,R_* are at the tips of the electrodes.

##### Repulsive Force between the Electrodes

To prevent an event where both electrodes would be pulled into the same position, or very close together, the electrodes are forced apart. The repulsive force is proportional to the inverse of distance between the electrode tips:(5)$$\begin{array}{c} F_{dd}\;=\;\frac{1}{\left\|dd\right\|}\cdot \frac{dd}{\left\|dd\right\|},\; \end{array}$$
where *dd* is the vector from tip of the electrode 1 toward tip of the electrode 2 (illustrated in Figure 5c). The application point of force *F_dd_* is at the tip of electrode 2 and the application point of force *−F_dd_* is at the tip of electrode 1.

##### Sum of Forces

The final forces *F*_*ele1*_ and *F*_*ele2*_, acting on the electrodes, and, therefore, effectively moving the electrodes to new positions, are a weighted sum of all acting forces:(6)$$\begin{array}{c} F_{ele1}\;=\;w_{1}\cdot F_{geo}+w_{2}\cdot \underset{i}{\sum }F_{1\;i,T}+w_{3}\cdot \underset{i}{\sum }F_{1\;i,R}+w_{4}\cdot \left(-F_{dd}\right),\;\; \end{array}$$
(7)$$\begin{array}{c} F_{ele2}\;=\;w_{1}\cdot F_{geo}+w_{2}\cdot \underset{i}{\sum }F_{2\;i,T}+w_{3}\cdot \underset{i}{\sum }F_{2\;i,R}+w_{4}\cdot F_{dd}.\;\; \end{array}$$

The concept of the forces acting on the electrodes is illustrated in Figure 5. After the new electrode positions are generated, the new voltage amplitude is calculated by multiplying the voltage-to-distance ratio, set to 1000 V/cm, and the distance between the electrode middle points, and rounded to 100 V. The electrode positions are changed in each iteration according to Equations (3)–(7). If the new electrode positions produce a decrease in CTV coverage, the algorithm reverts the positions to positions from the previous iteration and increases the applied voltage amplitude to 10% of the original value. The voltage-to-distance ratio is also increased accordingly and kept at an increased value through the rest of the optimization process.

#### 2.3.4. Termination and Handling of Errors

The optimization is terminated if 100% soft coverage of the CTV is achieved, if the change in CTV coverage between iterations is less than 0.1 percentage point (i.e., tolerance), or if either the maximum number of iterations (50 iterations), maximum allowed voltage (3000 V), or maximum allowed electric current (45 A) are reached.

In case a meshing error occurs after moving the electrodes (usually due to self-intersecting faces), the electrodes are displaced by 0.5 mm in a random direction. If the error persists after correction, the optimization is terminated, and the last computed results are saved.

In case the CTV is not completely covered after the optimization of electrode positions, the final step is to increase the applied voltage in steps of 100 V until either complete coverage of CTV or the maximum allowed voltage (3000 V) is reached.

#### 2.3.5. Algorithm Output

The output of the algorithm provides the final coordinates of the electrode’s entry point and the electrode tip, which correspond to the local coordinate system of the medical image used for initialization. The amplitude of the applied voltage and the expected current draw are also provided, as well as the expected coverage of the GTV and CTV with the selected electric field threshold (e.g., 400 V/cm). The total computation time and the number of iterations are also stored. All steps performed by the algorithm are stored in a log text file so that detailed information about the optimization process is available.

### 2.4. Full Factorial Experiment

The uncertainties in the algorithm stem from the four weights (*w1*, *w2*, *w3,* and *w4*) belonging to the forces in the final sum (Equations (6) and (7)). To understand the effects of the weights on the algorithm’s performance, a two-stage (low and high), four-factor, full-factorial experiment was conducted. Each weight was tested at a low and high stages and all combinations of the four weights were tested, resulting in a total of 16 groups, each with 108 replicates (models). The tested levels of parameters are listed in Table 2. The algorithm’s performance was measured with respect to computation time, number of iterations, and number of meshing errors. The goal was to achieve full coverage with minimum number of iterations, errors, and the shortest time. The measured data (computation time, number of iterations, and number of errors) in all 16 groups had the same non-normal distribution. Therefore, the Kruskal–Wallis test was used to statistically determine the effects of the weights. The Kruskal–Wallis test is a nonparametric version of the classic one-way ANOVA and uses ranks of the data rather than numerical values to calculate the test statistics.

## 3. Results

### 3.1. Full Factorial Experiment

The results of the full factorial experiment showed no significant effect of the algorithm weights on the performance (*p* = 0.917); therefore, no further optimization of the weights was performed. The algorithm’s performance is determined by three factors: the mean computation time, mean number of iterations, and number of meshing errors. All 16 weight groups were sorted and ranked (from 1–16) according to each performance factor. The three ranks were summed in each group to obtain the groups’ total scores, and the group with the lowest score was selected as the set of final algorithm weights. The results of the full factorial experiment along with the group scores are shown in Table 2.

### 3.2. Algorithm Performance

The performance of the algorithm was evaluated on realistic models of six lumbar (L1–L5) and six thoracic (T8–T12) vertebrae (created from patient images) with a total of 108 synthetic spherical tumors (created for simulation). The average computation time was 71 s (range: 17–253 s); the average and median number of iterations were 4.9 and 4.5, respectively (range: 1–15); and the average applied voltage amplitude was 2663 V (range: 1800–3000 V). The algorithm successfully completed the optimization in 103 models, while a meshing error occurred in five models. In 87/108 models, 100% coverage of CTV and GTV was achieved. In 17/108 models, a CTV coverage of more than 99% and a GTV coverage of 100% were achieved. In four models, the CTV coverage was greater than 94% and the GTV coverage was greater than 98%.

A meshing error occurred in 5/108 models, corresponding to 4.63%. All five errors occurred in the anterior-lateral tumor locations in the thoracic vertebral region with tumor radii of 7.5 mm or 10 mm. In all five models, 100% CTV and GTV coverage was achieved by an additional voltage increase after the last successful iteration. The termination criteria were complete CTV coverage in 70 models, maximum voltage in 22 models, minimum tolerance in 11 models, and meshing error in 5 models.

The dependence of the computation time and the number of iterations on the following variables was evaluated: vertebra type (lumbar/thoracic), tumor location, and tumor radius. Since the distribution of the data (computation time and number of iterations) does not follow a normal distribution, a nonparametric Kruskal–Wallis test was used. According to the test, the only variable with a statistically significant effect is the tumor radius (*p* < 0.01). This result is to be expected because only two electrodes were used, which makes it more difficult to cover larger tissue volumes. Table 3 shows the average computation time, number of iterations, and applied voltage for each tumor radius group.

## 4. Discussion

This study is one of the first attempts to use spatial information about the electric field distribution in tissues to optimize electrode positioning and pulse amplitude without using computationally intensive genetic algorithms. The algorithm is developed for the treatment of vertebral tumors using two needle electrodes inserted through the pedicles into the vertebral body. The concept is based on our previous studies on the electrochemotherapy of spinal metastases using a transpedicular approach [28,29]. The vertebral column is the most common site for bone metastases, with the incidence reaching up to 70% depending on the primary cancer type [32]. Electroporation offers several advantages over other treatment options, as it preserves the integrity of bone tissue, enables bone regeneration, and has low neural toxicity [33,34,35,36]. Studies have shown that bone metastases can be effectively treated with ECT with significant improvements in patients’ pain level and quality of life [29,33,37,38,39]. The transpedicular approach is a well-established technique used for tumor ablation, cement injection, and for the insertion of fixation screws in spinal fixation surgery [40,41,42]. Combining ECT with the technology used for transpedicular access could facilitate the introduction of the ECT or IRE ablation of vertebral tumors into clinical practice to further improve tumor control [28,43].

The treatment planning for electroporation-based treatments is still in the early stages of development [22,44,45]. The treatment plans are made prior to the procedure, and the electrode positions and voltage amplitudes are still primarily determined manually (by hand). This process usually requires several iterations where the operator changes the electrode positions between computations. After each iteration, the operator must visually inspect the electric field (usually as an overlay over the medical image), determine the potentially undertreated areas of the CTV, reposition the electrodes accordingly, and repeat the process. This approach requires a high level of expertise in the distribution of the electric field in inhomogeneous tissue and the impact of electrode positioning and pulse parameters [44,46].

When designing the algorithm, we followed the concept of the manual approach; however, the goal was to automate the process so as to require minimal operator input. The algorithm is modular, in a sense, and considers various requirements. For example, an appropriate distance between electrodes needs to be maintained at all times to avoid short circuits; it is also intuitive to move the electrodes toward the center of mass of the tumor or toward large regions of undertreated tissue. Furthermore, the concept of soft coverage is introduced, where we consider an electric field strength below the threshold in the CTV margin to be acceptable. The reason for this decision is that the thresholds currently used in treatment planning are only a very rough estimate. The threshold itself is a difficult property to determine. The determined threshold values are influenced by biological variability (small and large animals and humans), the condition of the tissue sample (in vivo vs. ex vivo), and the measurement method, among other factors. Therefore, there are a range of values in the literature, even for the same tissue type. Whether the tissue is electroporated also depends on the pulse protocol used, i.e., the number of pulses, pulse duration, and repetition rate. In addition, certain parts of the tissue may be cumulatively exposed to more pulses than other parts due to multiple pairs of active electrodes. Studies suggest that electroporation can occur at lower thresholds when the exposure time is increased with more and/or longer pulses [47,48,49,50].

The current implementation considered the technical limitations of the Cliniporator Vitae (IGEA S.p.A., Carpi, Italy), a commercially available pulse generator for electrochemotherapy. The allowed voltage amplitudes are 500–3000 V, rounded to 100 V, which correspond to the default step size of the generator, and is commonly used in clinical practice. With pulse generators, it is possible to set the voltage step size manually; therefore, this parameter was also included in the algorithm. However, decreasing the voltage step also increases the number of iterations required to obtain the optimal solution. The current limit is set at 45 A, which is 5 A lower than the pulse generator’s limit (50 A). Biological tissue is a very inhomogeneous material, and its actual electrical conductivity can vary significantly from the modelled values. Bones have low electrical conductivity compared to other tissues; therefore, the current is unlikely to reach the limit. However, in tissues with high conductivity, this may become a legitimate concern, and the algorithm’s limit should be set to lower values, since high current consumption will immediately terminate the pulse’s delivery.

The performance of the algorithm was evaluated on realistic vertebral models of the lower thoracic (T8–T12) and lumbar (L1–L5) segments (created from patient images) with inserted synthetic spherical tumor models of different sizes (created for simulation), resulting in 108 models in total. The results have shown that the algorithm performs successfully for different segments of the spine, different tumor sizes, and different locations within the vertebral body. A meshing error occurred in 4.63% of models; however, the algorithm still achieved complete coverage of the CTV and GTV. The most time-consuming step of the optimization process is the creation of the anatomical model based on medical imaging. However, this step is also required for any other treatment-planning concept. Once the model is completed, the operator only needs to select two points per pedicle in the patient’s medical image, and the treatment parameters are calculated within a few minutes. The average time to find a solution using the algorithm was 71 s (range: 17–253 s), and the average number of iterations was 4.9 (range: 1–15). This is a significant improvement over the solution-finding ability of a genetic algorithm, which requires at least 100 generations (equivalent to iterations in this case). It can also be assumed that optimization with the algorithm is faster than determining electrode positions by hand, since it essentially automates the same process, and a significantly lower level of expertise is required from the operator. It is worth noting that the process of image segmentation could also be automated to some degree, given the high contrast of bone tissue in CT imaging.

The algorithm returns the coordinates of the final electrode positions, which correspond to the coordinates of the medical image used for initialization. To produce an output that is useful to the surgeon performing the procedure, the electrode positions can be written into the DICOM files of the medical images by manipulating the brightness of the pixels in a manner that corresponds to the electrode positions. This way, the optimal positions can be inspected using medical image-viewing software that is available in hospitals. Alternatively, the electrode coordinates can be transformed into a set of morphological parameters that are commonly used to position transpedicular screws in spinal fixation surgery: the transversal angle, sagittal angle, distance from the sagittal plane (entry point), and insertion depth [51].

The main limitation of this study is the lack of validation towards realistic vertebral tumors. Before a treatment-planning workflow can be established, validation must be performed towards real clinical cases, either prospectively or retrospectively. A realistic tumor geometry could lead to some meshing issues that have not currently been encountered and would need further investigation. Another limitation is that the current implementation of the algorithm allows for the use of only two electrodes, which limits its use to tumors located mainly within the vertebral body, i.e., to the earlier stages of the disease.

In the future, additional electrodes could be added either in the same vertebra or in adjacent vertebrae using a similar concept, which would allow for the treatment of larger tumor volumes that are less well-contained (extend outside of the vertebral body). When adding new electrodes, the overlapping contributions of the different electrode pairs should be investigated and considered when calculating the soft coverage of the CTV. Furthermore, the size of the safety margin and the length of electrode exposure could be adjusted to the tumor size, the entry point could be shifted within the pedicle to allow for even better electrode positioning, and additional boundary conditions could be introduced, such as defining the minimum allowed distance to the center of the spinal cord, to ensure treatment safety, especially in IRE ablation, where high voltages are used and a substantial temperature rise around the electrodes is expected. [52,53,54]. Adapting the algorithm for other anatomic treatment sites, such as deep-seated soft tissue tumors, would require a slightly different approach to determining electrode placement, for example, in relation to the center of mass of the tumor. However, most of the concepts remain the same or require minimal modification.

## 5. Conclusions

This study introduces and presents an algorithm developed for the optimization of electrode positions (and pulse amplitudes) based on the spatial information of the electric field distribution within the target tissue. The algorithm is currently designed for the electrochemotherapy (and potentially irreversible electroporation ablation) of vertebral tumors using a transpedicular access but could be adapted to new anatomic sites in the future. The algorithm performed successfully for different segments of the spine, tumor sizes, and locations within the vertebral body. This study serves as a proof of concept that the electrode positions can be determined (semi)automatically based on the spatial information of the electric field distribution in the target tissue. The algorithm’s source code and all models of vertebral tumors created in this study are available in an online repository.

## Figures and Tables

**Figure 1 cancers-14-05412-f001:**
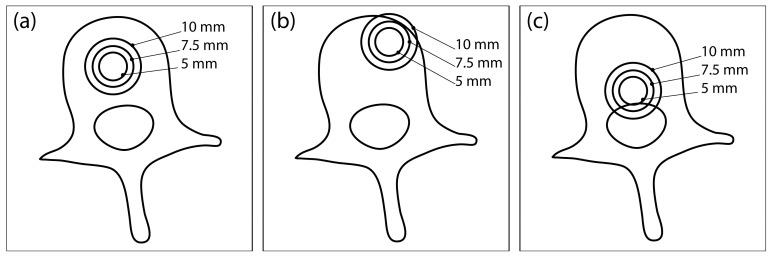
Three different tumor locations within the vertebral body: (**a**) central location, (**b**) anterior-lateral location, and (**c**) posterior-inferior location. At each location, the tumor is modelled with three different radii: 5 mm, 7.5 mm, and 10 mm. This illustration was created from an axial CT section of an L3 vertebra and does not show the actual geometry of the numerical model used for computation.

**Figure 2 cancers-14-05412-f002:**
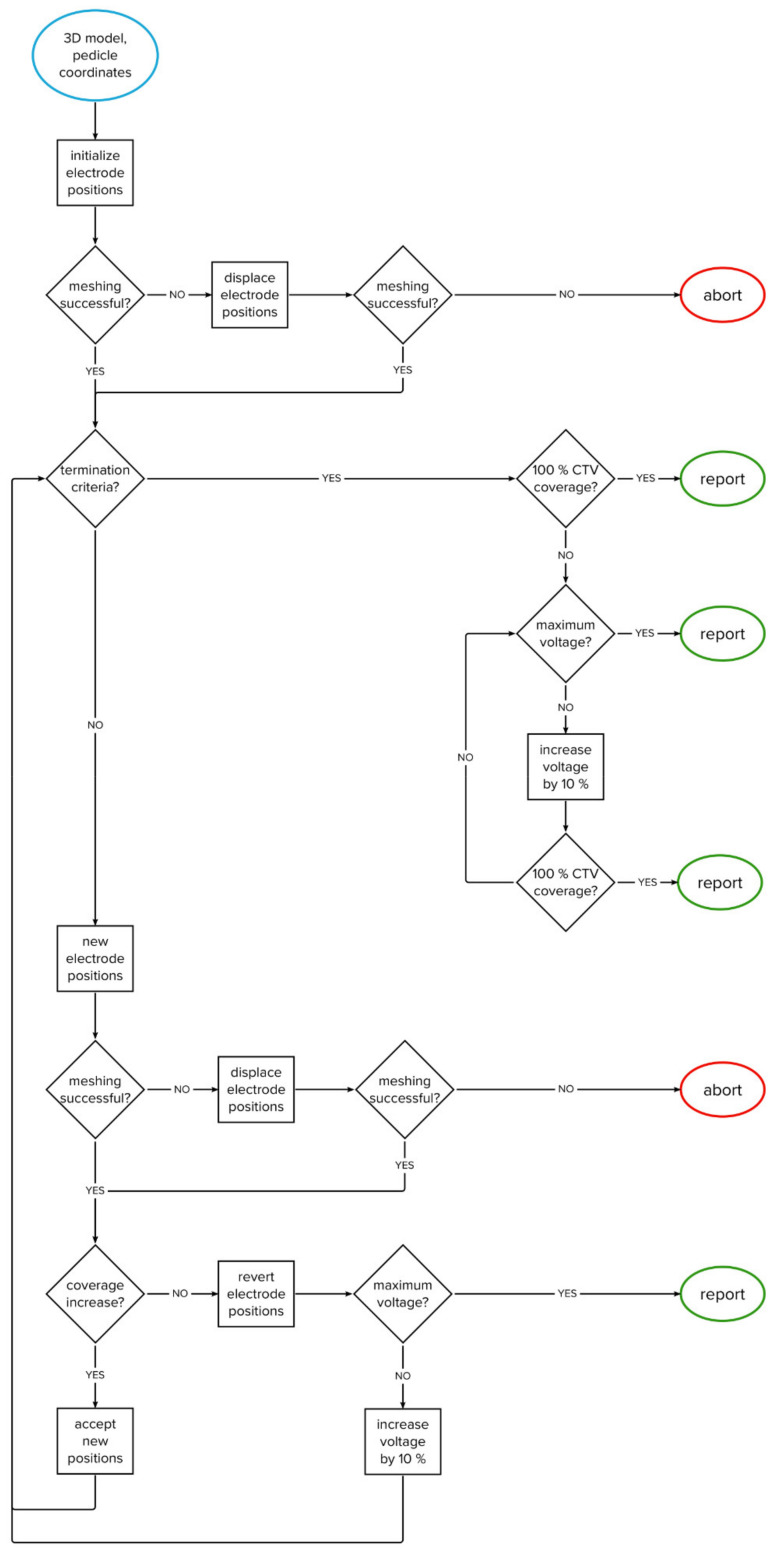
Flowchart of the algorithm.

**Figure 3 cancers-14-05412-f003:**
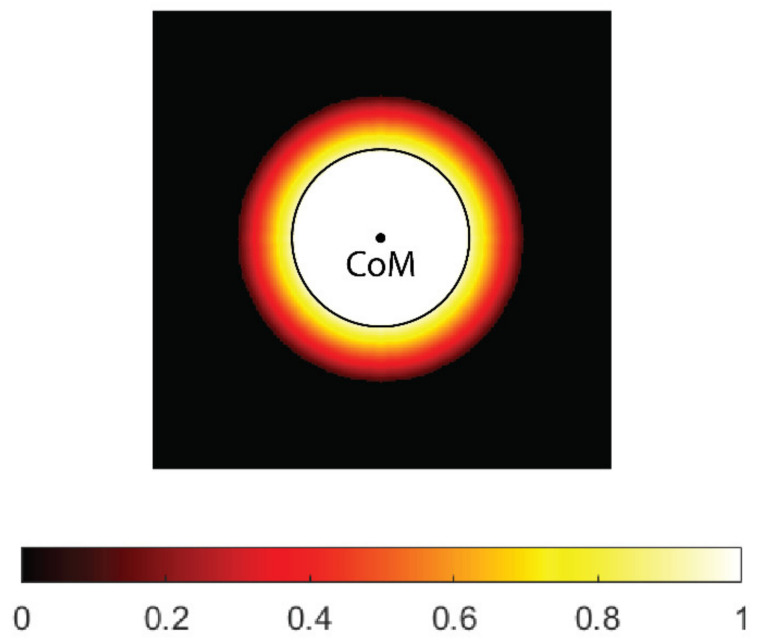
A weighting map of the clinical target volume (CTV). The contour (black) and center of mass (CoM) of the tumor gross volume (GTV, black circle) are shown. The weight uniformly decreases with distance from tumor border and reaches zero on the outer border of the CTV.

**Figure 4 cancers-14-05412-f004:**
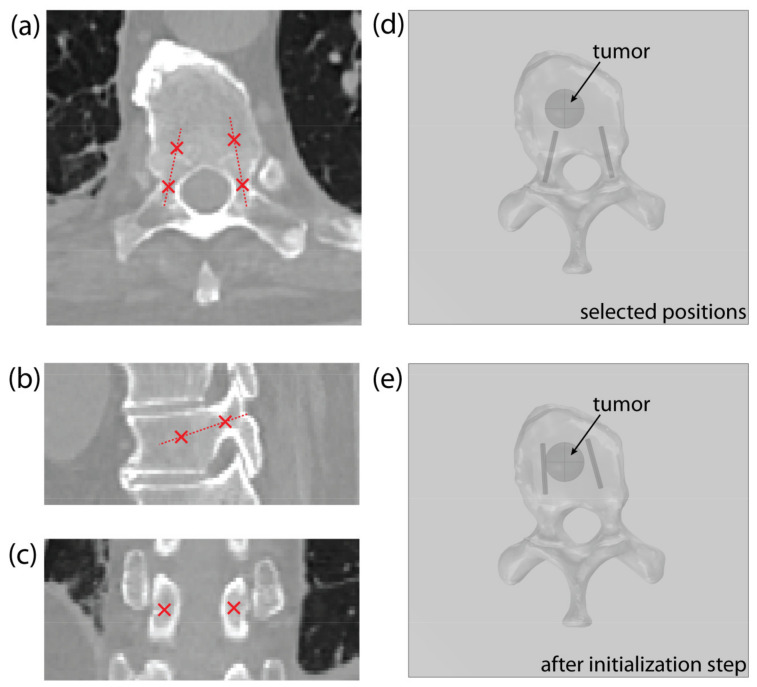
An example of point selection shown on the (**a**) axial, (**b**) sagittal, and (**c**) coronal CT slice of a thoracic vertebra. (**d**) An example of the model’s geometry in COMSOL Multiphysics, showing a thoracic vertebra and a spherical tumor, with starting electrode positions, obtained from the selected points. (**e**) Corrected electrode positions after algorithm initialization step.

**Figure 5 cancers-14-05412-f005:**
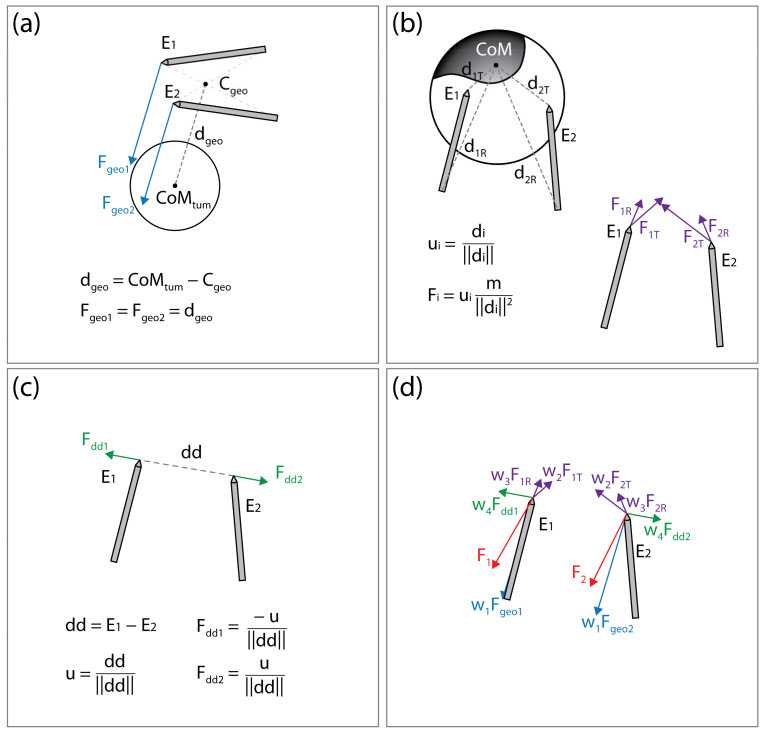
Illustration of the forces acting on the electrodes. (**a**) Attractive force (*F*_*geo1*_, *F*_*geo2*_) toward the tumor’s center of mass (*CoM_tum_*). (**b**) Attractive force (at the electrode tip, *F_iT_*, and rear *F_iR_*) toward the undertreated areas of the clinical target volume. (**c**) Repulsive force (*F*_*dd1*_, *F*_*dd2*_) maintaining appropriate distance between electrodes. (**d**) Final forces (*F*_*1*_ and *F*_*2*_) acting on the electrodes are a weighted sum of all forces. This figure is for illustration of the concept only; distances, vectors, and sums do not represent actual values.

**Table 1 cancers-14-05412-t001:** Electrical properties of modelled tissues and electrodes are taken from [28]. The surrounding tissue was assigned the properties of adipose tissue.

Tissue Property	Bone	Tumor	Surrounding Tissue	Electrodes
Initial electrical conductivity [S/m]	0.07	0.30	0.02	10^6^
Factor of electrical conductivity increase	2.9	2.8	3.0	-
Center of transition zone [V/cm]	600	600	300	-
Size of transition zone [V/cm]	400	400	400	-

**Table 2 cancers-14-05412-t002:** Results of the full factorial experiments for weights w1–w4.

w1	w2	w3	w4	Average Time (s)	Average Iterations (−)	Number of Errors (−)	Average Voltage (V)	Group Score (−)
0.7	0.1	0.02	5	81	5.1	3	2659	14
0.7	0.1	0.02	15	82	4.9	4	2687	11
0.7	0.1	0.14	5	84	5.5	2	2662	18
0.7	0.1	0.14	15	83	5.1	6	2682	20
0.7	0.7	0.02	5	84	5.6	4	2656	27
0.7	0.7	0.02	15	87	5.5	6	2689	28
0.7	0.7	0.14	5	86	5.7	2	2663	29
0.7	0.7	0.14	15	88	5.5	7	2693	34
1.3	0.1	0.02	5	71	4.9	5	2663	8 *
1.3	0.1	0.02	15	79	4.9	6	2694	12
1.3	0.1	0.14	5	72	4.9	8	2664	17
1.3	0.1	0.14	15	78	4.9	6	2693	11
1.3	0.7	0.02	5	83	5.5	8	2665	29
1.3	0.7	0.02	15	88	5.5	7	2690	34
1.3	0.7	0.14	5	85	5.6	7	2662	36
1.3	0.7	0.14	15	87	5.5	9	2693	37

The group with the lowest score (shadowed and indicated with *) has the best overall performance and was selected as the set of final algorithm weights.

**Table 3 cancers-14-05412-t003:** Tumor radius is the only variable affecting the algorithm’s performance. The average computation time, iterations, and applied voltage are shown for each modelled tumor radius.

Tumor Radius (mm)	Average Time (s)	Average Iterations (−)	Average Voltage (V)
5	31	1	2331
7.5	70	5	2667
10	113	9	2992

## Data Availability

The algorithm source codes and all 3D models of vertebral tumors created in this study are openly available in the FigShare repository at https://doi.org/10.6084/m9.figshare.21270111.v1 (accessed on 5 October 2022). No patient data or medical images are shared in the database.

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
