# Peer review of "Optimization of Transpedicular Electrode Insertion for Electroporation-Based Treatments of Vertebral Tumors"

_cancers, 2022, doi:10.3390/cancers14215412_

Round 1
Reviewer 1 Report
This study presents an algorithm developed for the optimization of electrode positions of vertebral tumors by using two electrodes.
For a successful electroporation treatment it is crucial to cover the whole target tissue volume with the appropriate electric field. To achieve this goal the planning of the best electrode position and voltage amplitude can be time consuming and requires great expertise.
According to the results of the study with the use of the developed algorithm on synthetic models of vertebral tumors the time of planning can be significantly reduced for different segments of the spine, different tumor sizes and locations within the vertebra.
In clinical practice however the treatment of tumors grown over the vertebral body could be necessary which means the need of additional electrodes (more than two) for the coverage of the whole clinical target volume.
In cases when more than two electrodes are necessary for the treatment the time of planning could be still reduced by using a similar concept to the presented algorithm?
Author Response
Q1: In cases when more than two electrodes are necessary for the treatment the time of planning could be still reduced by using a similar concept to the presented algorithm?
Re: Although this has not yet been supported by calculations, we strongly believe that the algorithm would be equally (if not more) beneficial in cases with more than two electrodes. With two electrodes it is relatively easy to mentally visualize the field distribution, but with, say, four electrodes it becomes more difficult and requires considerable operator expertise. Therefore, we often resort to simply testing different electrode configurations to see the result. Each actual electric field calculation takes about 10 s, and then the algorithm determines the areas not covered and moves the electrodes independently. So multiple iterations are performed in a very short period. Even though the average optimization time would probably be longer than 71 s, it would still be much faster than manually fine-tuning the electrode positions. From experience, we know that manual fine-tuning of the electrode configuration (in the model) with 4 electrodes can take 30 min or even more.

Reviewer 2 Report
The authors present a detailed algorithm for implementing electroporation as a treatment strategy for vertebral column tumors. The paper is well written and the therapeutic modality is an exciting new option. A few minor edits:
Figure 1: Does this represent a smoothed/merged image of a thoracic and lumbar body? Given the significant difference of upper thoracic and lower lumbar body anatomical shape, I think that the authors need to specify the constraints of this modeling. Specifically that the images are not applicable at the extreme of the T1-4 bodies.
In methods when you refer to a healthy block of tissue, was this theoretical or a cadaver specimen?
Lines 165-167: “For larger tumor radii, it may be difficult to cover the entire CTV. However, it is much 165 more important to cover areas closer to the tumor volume than areas at the outer edge of 166 the CTV where few or no tumor cells are expected; therefore, the so-called soft coverage 167 of the CTV is introduced. “ This is a problematic statement, as the focus of recurrence is typically at the margin of the tumor. If you are recommending a treatment modality that is up against surgical resection and or radiation, which can address tumor beyond the margins, I think that you need to justify this statement, remove it or qualify it.
Figure 4: I assume that the authors are using the small circle as the proposed tumor - it is difficult to visualize and I recommend labeling it more explicitly.
For algorithm performance (lines starting 312): Were these human patients or were they simulations? (Line 396 states that these were synthetic models but what does that mean?) IF human patients, can you please include the IRB or IRB exemption status of the study?
Author Response
Point 1: Figure 1: Does this represent a smoothed/merged image of a thoracic and lumbar body? Given the significant difference of upper thoracic and lower lumbar body anatomical shape, I think that the authors need to specify the constraints of this modeling. Specifically that the images are not applicable at the extreme of the T1-4 bodies.
Response: This is just an illustration created in Illustrator from axial images of an L3 vertebra; therefore, the smoothing is completely exaggerated. We have included the following explanation in the caption of Figure 1 (page 3, lines 106-107): “This illustration was created from an axial CT section of an L3 vertebra and does not show the actual geometry of the numerical model used for computation.”
We have also specified vertebral segments used in this study in the Methods (page 2), Results (page 10) and Discussion (page 12) sections.
Point 2: In methods when you refer to a healthy block of tissue, was this theoretical or a cadaver specimen?
Response: In this case, the properties of the block of the surrounding healthy tissue are theoretical – it is merely a background tissue, that is not particularly relevant for treatment inside the vertebral body. However, for the healthy bone tissue, we used conductivity data from our previous study (Cindric et al, 2018) in which we fitted the conductivity functions in the model based on current/voltage measurements on sheep vertebrae in vivo (Tschon et al, 2016). The surrounding tissue block was assigned the properties of adipose tissue. Although the exterior of the spine is covered with ligaments that have higher baseline conductivity than adipose tissue, we chose a tissue with lower conductivity as a precaution; to avoid overestimation of the treated area - a higher ratio between the conductivity of the tumor and the surrounding tissue makes it more difficult to cover the tumor (Miklavčič et al, 2010).
Cindrič H, Kos B, Tedesco G, Cadossi M, Gasbarrini A, Miklavčič D. Electrochemotherapy of Spinal Metastases Using Transpedicular Approach—A Numerical Feasibility Study. Technology in Cancer Research & Treatment. 2018;17. doi:10.1177/1533034618770253
Tschon M, Salamanna F, Ronchetti M, et al. Feasibility of Electroporation in Bone and in the Surrounding Clinically Relevant Structures: A Preclinical Investigation. Technology in Cancer Research & Treatment. 2016;15(6):737-748. doi:10.1177/1533034615604454
Miklavcic, D., Snoj, M., Zupanic, A. et al. Towards treatment planning and treatment of deep-seated solid tumors by electrochemotherapy. BioMed Eng OnLine 9, 10 (2010). doi:10.1186/1475-925X-9-10
Point 3: Lines 165-167: “For larger tumor radii, it may be difficult to cover the entire CTV. However, it is much more important to cover areas closer to the tumor volume than areas at the outer edge of the CTV where few or no tumor cells are expected; therefore, the so-called soft coverage of the CTV is introduced.” This is a problematic statement, as the focus of recurrence is typically at the margin of the tumor. If you are recommending a treatment modality that is up against surgical resection and or radiation, which can address tumor beyond the margins, I think that you need to justify this statement, remove it or qualify it.
Response: We thank the reviewer for bringing this point to our attention. We agree that the statement can be easily misinterpreted and have therefore removed it. The following explanation has been added to Section 2.3.1 on page 4:
“For safety reasons, the thresholds for electroporation used in practice are generally quite high, and sometimes it may be difficult to cover the entire CTV for larger tumor radii. However, if the safety margin is taken into account, very few or no cells are expected at the outer edge of the CTV; therefore, the so-called soft coverage of the CTV is introduced in this study, in which the threshold for electroporation at the outer edge of the CTV is not strictly enforced. /…/ The weighting map can be easily adapted to the tumor type. For example, in metastatic tumors, a higher weight (e.g., 0.5) can be assigned to the outer edge of the CTV to provide additional safety. /.../ The margin can be easily adjusted (e.g., 10 mm) for different tumor types.”
Point 4: Figure 4: I assume that the authors are using the small circle as the proposed tumor - it is difficult to visualize and I recommend labeling it more explicitly.
Response: The visibility of the image was improved, and the tumor was labeled accordingly.
Point 5: For algorithm performance (lines starting 312): Were these human patients or were they simulations? (Line 396 states that these were synthetic models but what does that mean?) IF human patients, can you please include the IRB or IRB exemption status of the study?
Response: Vertebrae were segmented from real patient images (as part of a retrospective validation of a treatment planning framework we are developing-not part of the current study). For the purposes of this study, spherical tumors were then added to the vertebral models. By the term "synthetic models," we refer to the spherical tumor models that were added to the realistic vertebral models.
Lines 313-316 on page 10 have been reworded for clarification and now read:
“The performance of the algorithm was evaluated on realistic models of 6 lumbar (L1-L5) and 6 thoracic (T8-T12) vertebrae (created from patient images) with a total of 108 synthetic spherical tumors (created for simulation).”
Lines 398-401 on page 12 have been reworded for clarification and now read:
“The performance of the algorithm was evaluated on realistic vertebral models of the lower thoracic (T8-T12) and lumbar (L1-L5) segments (created from patient images) with inserted synthetic spherical tumor models of different size (created for simulation), resulting in 108 models in total.”
The IRB statement has been included in the manuscript (page 13).
